# Vibration and Damping Analysis of Pipeline System Based on Partially Piezoelectric Active Constrained Layer Damping Treatment

**DOI:** 10.3390/ma14051209

**Published:** 2021-03-04

**Authors:** Yuanlin Zhang, Xuefeng Liu, Weichong Rong, Peixin Gao, Tao Yu, Huawei Han, Langjun Xu

**Affiliations:** 1School of Electromechanical and Automotive Engineering, Yantai University, Yantai 264005, China; zhanglintf1207@outlook.com (Y.Z.); liuxuefeng1121@163.com (X.L.); Rongweichong1@163.com (W.R.); 2Yantai CIMC Raffles Offshore Limited, Yantai 264670, China; Huawei.han@cimc-raffles.com (H.H.); langjun.xu@cimc-raffles.com (L.X.)

**Keywords:** pipeline, finite element method, piezoelectric materials, active control, viscoelastic materials

## Abstract

Pipelines work in serious vibration environments caused by mechanical-based excitation, and it is thus challenging to put forward effective methods to reduce the vibration of pipelines. The common vibration control technique mainly uses the installation of dampers, constrained layer damping materials, and an optimized layout to control the vibration of pipelines. However, the passive damping treatment has little influence on the low frequency range of a pipeline system. Active control technology can obtain a remarkable damping effect. An active constrained layer damping (ACLD) system with piezoelectric materials is proposed in this paper. This paper aims to investigate the vibration and damping effect of ACLD pipeline under fixed support. The finite element method is employed to establish the motion equations of the ACLD pipeline. The effect of the thickness and elastic modulus of the viscoelastic layer, the laying position, and the coverage of ACLD patch, and the voltage of the piezoelectric material are all considered. The results show that the best damping performance can be obtained by selecting appropriate control parameters, and it can provide effective design guidance for active vibration control of a pipeline system.

## 1. Introduction

Pipeline systems are important transmission channels of energy and material in practical engineering applications. Pipelines are connected to each component and need to bear the highest pressure of the system. A pipeline system is mainly composed of a pipeline, clamps, valves, and so on. A pipeline system needs more manpower and material force in its maintenance because of the complex crisscrossed layout and the narrow space between the pipelines. When a pipeline works, it is excited by the pulsating excitation from the pump source and the foundation excitation of the structure. The pipeline resonates when the external excitation frequency is close to the modal frequency of the pipeline. At this point, the pipeline will produce large displacement, resulting in large stress. The action of long-term high stress will cause damage to and the failure of the pipeline. The forms of pipeline failure generally include pipeline collision, wear, burst, noise, and so on. With the development of the modernization of related industries, considering the safety and economy of pipeline systems, the industry has put forward higher requirements for the stability and reliability of pipeline systems. Composite materials are extensively used as key components in aerospace and mechanical engineering sectors owing to their high specific strength and specific stiffness [1]. The most important problem to improve the stability and reliability of pipeline systems is the vibration of pipelines. Therefore, it is necessary for us to control the vibration of a pipeline using a more effective method.

Researchers have done a lot of research on the vibration of pipelines. Pipeline vibration is mainly due to the coupling effect of the liquid in the pipeline and the coupling vibration between the components in the hydraulic system. According to this vibration mechanism, the most commonly used vibration reduction methods are passive constrained layer damping control (PCLD) and active constrained layer damping control (ACLD). The passive control method pastes some inertia accessories on the structure to achieve the effect of vibration reduction [2]. Passive constrained layer damping includes a viscoelastic material (VEM) layer and a constrained layer, and the damping capacity is controlled by changing the shear strain of viscoelastic materials [3]. Gao et al. [4] established a finite element model of a passive constrained layer damping pipeline under elastic boundary support, and studied the influence of key parameters such as support stiffness, fluid velocity and pressure, the thickness and the elastic modulus of VEM on pipeline vibration. The characteristics of PCLD include the structure being simple and without external energy input. However, the effect of PCLD treatment on low-frequency and large-amplitude vibration is not obvious. Compared with the PCLD technology, the ACLD uses a piezoelectric constraining layer instead of the conventional constraining layer of PCLD. ACLD has a stronger adaptive ability than PCLD. ACLD damping includes the advantages of the ability of passive damping in the high-frequency range and the control active damping in the low-frequency range [5]. Zhao et al. [6] adopted active constrained layer damping control for an aero pipeline, and analyzed the transmission of driving force under the harmonic voltage, the influence of control parameters and the structural parameters on pipeline vibration. The research methods of pipeline vibration have matured, and include the Galerkin-type method, the transfer matrix method, the characteristic line method, and the finite element method. Kheirl et al. [7] used the Galerkin-type method to analyze the stability of a pipeline conveying fluid under flexible support, and verified the simplified motion equation. Li et al. [8] obtained the analytical solution in the time domain by using the characteristic line method, and the effectiveness of the method was verified by their experiments. Gao et al. [9] established the three-dimensional finite element modal, and discussed the influence of different voltages, control parameters, and structural parameters on pipeline vibration.

ACLD technology has been used widely in many fields of engineering. As early as 1993, Agnes [10] proposed the concept of active constrained layer damping. Subsequently, Baz [11], Shen [12], Lesieutre [13], and Liao [14] optimized and extended the concept and treatment method of active constrained layer damping. The finite element model and vibration control of beams with active constrained layer damping was proposed by Balamurugan [15]. Sonti [16] proposed the application of the active constraint layer technique in a shell element model. In recent years, ACLD treatment has been developing steadily. With the development of ACLD treatment, many excellent research results about ACLD have been published at home and abroad [17,18,19,20,21].

The parameters of ACLD have a great effect on vibration reduction. Therefore, it is necessary to study the influence of the parameters of ACLD to obtain the optimal value of treatment effect. The paper [22] investigated the performances of four types of hybrid active-passive constrained layer damping treatment. The changes in parameters between active, passive, and hybrid treatment were studied in paper [23]. In this paper, the influence of the damping length of the active constrained layer and the thickness of the viscoelastic material on the vibration of beam was obtained. The paper [24] studied the influence of an ACLD patch for the vibration attenuation of a thin truncated conical shell. Considering the finite element model of an ACLD plate using the modal strain energy method, the location of ACLD was optimized [25]. In paper [26], open-loop and closed-loop dynamic models of active and passive constrained layer damping beams were established, and the effect of key parameters such as control gain, VEM thickness, ACLD axial coverage, and position change on the loss factors of beam system were studied in detail. Based on Timoshenko beam theory and the Hamilton principle, the vibration control equation of a beam with distributed internal viscous damping was established in paper [27], and the frequency equation of the beam was gained by using transfer matrix method. There are some papers [28,29,30,31] which carried out a further exploration in improving and optimizing the active passive constrained layer damping structure. The purpose of these papers was to select the optimal parameters of an ACLD structure through the research and simulation, so that the vibration performance of the structure can reach the optimal state.

As mentioned above, the authors find that regarding active vibration control, most scholars paid attention to the application of beams and shells, while there are few studies on pipeline systems. Therefore, this is the first time that the vibration of an ACLD pipeline under fixed boundary supports is investigated. Due to the particularity of the pipeline structure and the complexity of the excitation, research on pipeline vibration is very important. So, this paper establishes the finite element model and deduces the dynamic equation of the pipeline. The influence of structural parameters and control parameters on pipeline vibration is discussed.

## 2. Finite Element Model of the Pipeline

The boundary condition is very important in the process of establishing the finite element model. Figure 1 shows a pipe model with a partially covered ACLD layer under fixed boundary support, and a piezoelectric sensor is attached to the pipeline. *x*0 is the distance from ACLD to the left fixed support, and *x*1 is the covering length of ACLD in the *z*-axis direction of the pipeline. The length of *x*1 is symmetrical for both sides relative to the middle position of the pipeline.

Based on the following assumptions, the finite element model of pipeline is established.

The rotation and shear deformation of piezoelectric layer and base pipeline are not considered;The viscoelastic layer has a transverse variable, but the normal stress is negligible;The pipeline, viscoelastic damping layer, and piezoelectric layer are all in the linear elastic range;There is no relative slip between the layers and the interfaces have a perfect continuity;The potential distribution of piezoelectric layer is linear along the thickness direction;The complex shear modulus of viscoelastic material is independent of temperature.

### 2.1. Displacement Fields

Figure 2 plots the geometry and deformation of the ACLD element of the pipeline. Based on Timoshenko beam theory, the dynamic equations of viscoelastic layer are established. The elastic layer equations are established with the Euler Bernoulli beam theory. $t_{b}$, $t_{v}$, and $t_{p}$ are the thickness of the pipeline, viscoelastic layer, and piezoelectric layer, respectively. The neutral axis displacement of the piezoelectric layer and baseline layer are expressed by $u_{p}$ and $u_{b}$, respectively. w denotes the transverse displacement, while $r_{b}$, $r_{v}$, and $r_{p}$ denote the radius of the base pipe, the viscoelastic layer, and the piezoelectric layer, respectively. The rotation of the viscoelastic layer from normal to the mid-plane is γ. θ is the coverage angle of ACLD along the pipeline circumference.

From the strain field and displacement field, the constitutive relation of ACLD pipe system can be obtained. For *i* = b, p, the displacement and strain of the pipeline and the piezoelectric constraining layer can be gained from the neutral axial displacement and the transverse displacement.
(1)$$U_{i}=u_{i}-r\sin\theta \frac{\partial w}{\partial x}\;\;\;\;\;\;\;\;\;\;\;\;\;\;\;\;r_{i}-h_{i}/2\leq r\leq r_{i}+h_{i}/2$$
(2)$$\varepsilon_{i}=\frac{\partial u_{i}}{\partial x}-r\sin\theta \frac{\partial^{2}w}{\partial x^{2}}$$
where $U_{i}$ and $\varepsilon_{i}$ represent the displacement and the normal strain. The displacement and strain of viscoelastic layer can be written as:(3)$$U_{v}=u_{v}\;+(r-r_{p}+t_{p}/2)\left(\gamma -\frac{\partial w}{\partial x}\right)\;\;\;\;\;\;\;\;\;\;\;\;\;\;r_{p}-t_{p}/2\leq r\leq r_{p}+t_{p}/2$$
(4)$$\varepsilon_{v}=\frac{\partial u_{v}}{\partial x}+(r-r_{p}+t_{p}/2)\;\left(\frac{\partial \gamma }{\partial x}+\frac{\partial^{2}w}{\partial x^{2}}\right)$$

Due to the perfect continuity, the neutral axis displacement of the pipeline and piezoelectric layer can be expressed as:(5)$$u_{p}\;=u_{v}+\frac{t_{v}}{2}\left(\gamma -\frac{\partial w}{\partial x}\;\right)-\frac{t_{p}}{2}\frac{\partial w}{\partial x}$$
(6)$$u_{b}\;=u_{v}-\frac{t_{v}}{2}\left(\gamma -\frac{\partial w}{\partial x}\;\right)+\frac{t_{b}}{2}\frac{\partial w}{\partial x}$$

From Equations (5) and (6), the neutral axial displacement and shear strain of the viscoelastic layer are represented as:(7)$$u_{v}=\frac{1}{2}\left[\left(u_{p}+u_{b}\right)+\frac{t_{p}-t_{b}}{2}\frac{\partial w}{\partial x}\right]$$
(8)$$\gamma =\frac{1}{t_{v}}\left[u_{p}-u_{b}+(t_{v}+\frac{t_{b}+t_{p}}{2})\frac{\partial w}{\partial x}\right]$$

### 2.2. Shape Functions

In this paper, the element with two nodes (*i* and *j*) is defined. The local nodal displacement of ACLD pipeline element can be expressed as:(9)$$\left\{q^{e}\right\}={\left[w_{i},{w^{\prime }}_{i}{}_{,},u_{pi},u_{bi},w_{j},{w^{\prime }}_{j},u_{pj},u_{bj}\right]}^{T}$$
where w and $w^{\prime }$ are, respectively, the transverse displacement and the slope. $u_{b}$ and $u_{p}$ are the axial displacement of pipeline and piezoelectric layer, respectively.

$N_{w}(x)$ is the transverse shape function, $N_{p}(x)$ is the axis shape function of the constrained layer, while $N_{b}(x)$ is the axis shape function of the base pipeline.
(10)$$\left\{\begin{array}{c} w(x)\\ u_{p}(x)\\ u_{b}(x) \end{array}\right\}=\left[\begin{array}{c} N_{w}(x)\\ N_{p}(x)\\ N_{b}(x) \end{array}\right]\left\{q^{e}\right\}$$
where
(11)$$N_{w}(x)=\left[\begin{array}{llllllll} 1-3{\left(\frac{x}{l_{0}}\right)}^{2}+2{\left(\frac{x}{l_{0}}\right)}^{3} & x-2\frac{x^{2}}{l_{0}}+\frac{x^{3}}{l_{0}^{2}} & 0 & 0 & 3{\left(\frac{x}{l_{0}}\right)}^{2}-2{\left(\frac{x}{l_{0}}\right)}^{3} & -\frac{x^{2}}{l_{0}}+\frac{x^{3}}{l_{0}^{2}} & 0 & 0 \end{array}\right]$$
(12)$$N_{p}(x)=\left[\begin{array}{llllllll} 0 & 0 & 1-\frac{x}{l_{0}} & 0 & 0 & 0 & \frac{x}{l_{0}} & 0 \end{array}\right]$$
(13)$$N_{b}(x)=\left[\begin{array}{llllllll} 0 & 0 & 0 & 1-\frac{x}{l_{0}} & 0 & 0 & 0 & \frac{x}{l_{0}} \end{array}\right]$$

The shear strain and axial displacement of viscoelastic layer can be expressed as:(14)$$\gamma =\left[N_{\gamma v}\right]q^{e}$$
(15)$$u_{v}=\left[N_{uv}\right]q^{e}$$
where
(16)$$\begin{array}{l} \left[N_{rv}\right]=[\begin{array}{cccc} 6t\left(-\frac{x}{l_{0}^{2}}+\frac{x^{2}}{l_{0}^{3}}\right) & t\left(1-4\left(\frac{x}{l_{0}}\right)+3{\left(\frac{x}{l_{0}}\right)}^{2}\right) & 1-\frac{x}{l_{0}} & -1+\frac{x}{l_{0}} \end{array}\\ \begin{array}{cccc} \;\;\;\;\;\;\;\;\;\;\;\;\;\;\;\;\;\;\;\;\;\;\;\;\;\;\;\;\;\;\;\;\;\;\;\;\;\;\;\;\;6t\left(\frac{x}{l_{0}^{2}}-\frac{x^{2}}{l_{0}^{3}}\right) & t\left(-2\left(\frac{x}{l_{0}}\right)+3{\left(\frac{x}{l_{0}}\right)}^{2}\right) & \frac{x}{l_{0}} & -\frac{x}{l_{0}} \end{array}] \end{array}$$
(17)$$\begin{array}{l} \left[N_{uv}\right]=[\begin{array}{cccc} \frac{3\left(t_{p}-t_{b}\right)}{2}\left(-\frac{x}{l_{0}^{2}}+\frac{x^{2}}{l_{0}^{3}}\right) & \frac{t_{p}-t_{b}}{4}\left(1-4\left(\frac{x}{l_{0}}\right)+3{\left(\frac{x}{l_{0}}\right)}^{2}\right) & \frac{1}{2}\left(1-\frac{x}{l_{0}}\right) & \frac{1}{2}\left(1-\frac{x}{l_{0}}\right) \end{array}\\ \;\;\;\;\;\;\;\;\;\;\;\;\;\;\;\;\;\;\;\;\;\;\;\;\;\;\;\;\;\;\;\;\;\;\;\;\;\;\;\begin{array}{cccc} \;\;\;\frac{3\left(t_{p}-t_{b}\right)}{2}\left(\frac{x}{l_{0}^{2}}-\frac{x^{2}}{l_{0}^{3}}\right) & \frac{t_{p}-t_{b}}{4}(-2\left(\frac{x}{l_{0}}\right)+3{\left(\frac{x}{l_{0}}\right)}^{2}) & \frac{x}{2l_{0}} & \frac{x}{2l_{0}} \end{array}] \end{array}$$
where $t=t_{v}+\frac{\left(t_{b}+t_{p}\right)}{2}$.

### 2.3. Energy Expressions

The associated mass and stiffness matrices of the ACLD pipeline are determined from the energy expression in this section.

#### 2.3.1. Base Pipeline Layer

The potential energy of the pipe due to bending can be written as:(18)$$\frac{1}{2}E_{b}I_{b}\int_{0}^{l_{0}}{\left(\frac{\partial^{2}w}{\partial x^{2}}\right)}^{2}dx=\frac{1}{2}E_{b}I_{b}{\left\{q^{e}\right\}}^{T}\int_{0}^{l_{0}}N_{w}{\left(x\right)}_{xx}^{T}N_{w}{\left(x\right)}_{xx}dx\left\{q^{e}\right\}=\frac{1}{2}{\left\{q^{e}\right\}}^{T}K_{bb}^{pe}\left\{q^{e}\right\}$$
(19)$$K_{bb}^{pe}=E_{b}I_{b}\int_{0}^{l_{0}}N_{w}{\left(x\right)}_{xx}^{T}N_{w}{\left(x\right)}_{xx}dx$$

The potential energy of the pipeline due to extension can be represented as:(20)$$\frac{1}{2}E_{b}S_{b}\int_{0}^{l_{0}}{\left(\frac{\partial u_{b}}{\partial x}\right)}^{2}dx=\frac{1}{2}E_{b}S_{b}{\left\{q^{e}\right\}}^{T}\int_{0}^{l_{0}}N_{b}{\left(x\right)}_{x}^{T}N_{b}{\left(x\right)}_{x}dx\left\{q^{e}\right\}=\frac{1}{2}{\left\{q^{e}\right\}}^{T}K_{bp}^{pe}\left\{q^{e}\right\}$$
(21)$$K_{bp}^{pe}=E_{b}S_{b}\int_{0}^{l_{0}}N_{b}{\left(x\right)}_{x}^{T}N_{b}{\left(x\right)}_{x}dx$$
where $E_{b}$ and $S_{b}$ are the Young’s modulus and cross-section area of the base pipeline, and $I_{b}$ is the moment of inertia of the pipe. $K_{bb}^{pe}$ and $K_{bp}^{pe}$ are the element stiffness matrices of the pipeline.

The kinetic energy of the pipeline associated with transverse motion can be written as:(22)$$\frac{1}{2}\rho_{b}S_{b}\int_{0}^{l_{0}}{\left(\frac{\partial w}{\partial t}\right)}^{2}dx=\frac{1}{2}\rho_{b}S_{b}{\left\{q^{e}\right\}}^{T}\int_{0}^{l_{0}}N_{w}{\left(x\right)}^{T}N_{w}(x)dx\left\{q^{e}\right\}=\frac{1}{2}{\left\{q^{e}\right\}}^{T}M_{bb}^{pe}\left\{q^{e}\right\}$$
(23)$$M_{bb}^{pe}=\rho_{b}S_{b}\int_{0}^{l_{0}}N_{w}{\left(x\right)}^{T}N_{w}(x)dx$$

The kinetic energy of the pipeline associated with axial motion can be expressed as:(24)$$\frac{1}{2}\rho_{b}S_{b}\int_{0}^{l_{0}}{\left(\frac{\partial u_{b}}{\partial t}\right)}^{2}dx=\frac{1}{2}\rho_{b}S_{b}{\left\{q^{e}\right\}}^{T}\int_{0}^{l_{0}}N_{b}{\left(x\right)}^{T}N_{b}(x)dx\left\{q^{e}\right\}=\frac{1}{2}{\left\{q^{e}\right\}}^{T}M_{bp}^{pe}\left\{q^{e}\right\}$$
(25)$$M_{bp}^{pe}=\rho_{b}S_{b}\int_{0}^{l_{0}}N_{b}{\left(x\right)}^{T}N_{b}(x)dx$$
where $\rho_{b}$ denotes the pipeline density. $M_{bb}^{pe}$ and $M_{bp}^{pe}$ are the element mass matrices for the baseline.

#### 2.3.2. Viscoelastic Layer

The potential energy of the viscoelastic layer associated with the shear strain can be written as:(26)$$\frac{1}{2}G_{v}S_{v}\int_{0}^{l_{0}}\lambda^{2}dx=\frac{1}{2}G_{v}S_{v}{\left\{q^{e}\right\}}^{T}\int_{0}^{l_{0}}N_{\gamma v}{\left(x\right)}^{T}N_{\gamma v}(x)dx\left\{q^{e}\right\}=\frac{1}{2}{\left\{q^{e}\right\}}^{T}K_{\gamma v}^{pe}\left\{q^{e}\right\}$$
(27)$$K_{\gamma v}^{pe}=G_{v}S_{v}\int_{0}^{l_{0}}N_{\gamma v}{\left(x\right)}^{T}N_{\gamma v}(x)dx$$

The kinetic energy of the viscoelastic layer in transverse direction motion can be represented as:(28)$$\frac{1}{2}\rho_{v}S_{v}\int_{0}^{l_{0}}{\left(\frac{\partial w}{\partial t}\right)}^{2}dx=\frac{1}{2}\rho_{v}S_{v}{\left\{q^{e}\right\}}^{T}\int_{0}^{l_{0}}N_{w}{\left(x\right)}^{T}N_{w}(x)dx\left\{q^{e}\right\}=\frac{1}{2}{\left\{q^{e}\right\}}^{T}M_{vb}^{pe}\left\{q^{e}\right\}$$
(29)$$M_{vb}^{pe}=\rho_{v}S_{v}\int_{0}^{l_{0}}N_{w}{\left(x\right)}^{T}N_{w}(x)dx$$

The kinetic energy of the viscoelastic layer associated with the axial direction motion can be written as:(30)$$\frac{1}{2}\rho_{v}S_{v}\int_{0}^{l_{0}}{\left(\frac{\partial u_{v}}{\partial t}\right)}^{2}dx=\frac{1}{2}\rho_{v}S_{v}{\left\{q^{e}\right\}}^{T}\int_{0}^{l_{0}}N_{uv}{\left(x\right)}^{T}N_{uv}(x)dx\left\{q^{e}\right\}=\frac{1}{2}{\left\{q^{e}\right\}}^{T}M_{vp}^{pe}\left\{q^{e}\right\}$$
(31)$$M_{vp}^{pe}=\rho_{v}S_{v}\int_{0}^{l_{0}}N_{uv}{\left(x\right)}^{T}N_{uv}(x)dx$$
where $G_{v}$ denotes the complex shear modulus of the viscoelastic layer, while $\rho_{v}$ and $S_{v}$ are, respectively, the density and cross-section area of the viscoelastic layer. $K_{\gamma v}^{pe}$ is the element stiffness matrix of the viscoelastic layer. $M_{vb}^{pe}$ and $M_{vp}^{pe}$ are the element mass matrices for the viscoelastic layer.

#### 2.3.3. Piezoelectric Layer

The process of extracting the piezoelectric layer is similar to the base pipeline.

The kinetic energy of piezoelectric layer in transverse direction motion can be written as
(32)$$M_{pb}^{pe}=\rho_{p}S_{p}\int_{0}^{l_{0}}N_{w}{\left(x\right)}^{T}N_{w}(x)dx$$

The kinetic energy of the piezoelectric layer associated with the axial motion is
(33)$$M_{pp}^{pe}=\rho_{p}S_{p}\int_{0}^{l_{0}}N_{p}{\left(x\right)}^{T}N_{p}(x)dx$$

The potential energy of the piezoelectric layer due to bending can be represented as
(34)$$K_{pb}^{pe}=E_{p}I_{p}\int_{0}^{l_{0}}N_{w}{\left(x\right)}_{xx}^{T}N_{w}{\left(x\right)}_{xx}dx$$

The potential energy of the piezoelectric layer due to extension can be written as
(35)$$K_{pp}^{pe}=E_{p}S_{p}\int_{0}^{l_{0}}N_{p}{\left(x\right)}_{x}^{T}N_{p}{\left(x\right)}_{x}dx$$

In the case of one dimension, the constitutive relation of piezoelectric materials can be simplified into four different kinds of piezoelectric equations according to different independent variables, among which the following type is the most commonly used.
(36)$$\left[\begin{array}{l} S\\ D \end{array}\right]=\left[\begin{array}{cc} S_{11}^{E} & d_{31}\\ d_{31} & \varepsilon_{33}^{\sigma } \end{array}\right]\left[\begin{array}{l} \sigma \\ E \end{array}\right]$$
where S is the mechanical strain, D is the electric displacement, σ is the mechanical stress, and *E* is the electric field intensity. The elastic compliance constant is expressed by $S_{11}^{E}$, $\varepsilon_{33}^{\sigma }$ is the dielectric constant, and $d_{31}$ denotes the piezoelectric constant. The stress–strain relationship can be gained by the constitutive relation
(37)$$\tau =E_{p}(S-d_{31}E)$$
where
(38)$$E_{p}=\frac{1}{S_{11}^{E}};E=\frac{V_{p}(t)}{t_{p}}$$

The virtual work done by the induced strain (force) in voltage amplifier is
(39)$$\delta W_{p}=2\int_{0}^{l_{0}}\int_{0}^{2\pi }E_{p}d_{31}V_{p}(t)\delta (\frac{\partial u_{p}}{\partial x})d\theta dx={\left[\delta q(e)\right]}^{T}\{f_{p}\}$$
(40)$$\{f_{p}\}=\int_{0}^{2\pi }E_{c}d_{31}V_{p}(t){\left[N_{p}(x)\right]}^{T}d\theta$$
where $V_{p}(t)$ denotes the voltage applied to the piezoelectric layer.
(41)$$V_{p}(t)=-K_{d}{\dot{V}}_{S}$$
where $K_{d}$ is the derivative control gain. $V_{S}$ is the sensor output voltage, and it can be represented as:(42)$$V_{S}=-\left[\frac{k_{{}_{31}}^{2}H_{0}}{g_{31}C}\right]\int_{0}^{l_{0}}\int_{0}^{2\pi }\frac{\partial^{2}w}{\partial x^{2}}d\theta dx$$
where $k_{31}$ is the electromechanical coupling factor, and $H_{0}$ is the distance between neutral axis of the pipeline and piezoelectric sensor layer. $g_{31}$ denotes the piezoelectric voltage constant, while C is the capacitance of the sensor, and it can be expressed as
(43)$$C=8.854\times 10^{-12}A\frac{\varepsilon_{1}}{t_{p}}$$
where A is the sensor surface area, and $\varepsilon_{1}$ is the dimensionless dielectric constant.

### 2.4. Finite Element Implementation

According to the description in the above section, the final stiffness and mass matrix of ACLD pipeline can be written as
(44)$$\{\begin{array}{l} K^{e}=K_{bp}^{pe}+K_{bb}^{pe}+K_{pp}^{pe}+K_{pb}^{pe}+K_{\gamma v}^{pe}\\ M^{e}=M_{bp}^{pe}+M_{bb}^{pe}+M_{pp}^{pe}+M_{pb}^{pe}+M_{vp}^{pe}+M_{vb}^{pe} \end{array}$$

The virtual work done by an external force can be expressed as
(45)$$\delta W_{ex}=\int_{0}^{l_{0}}f_{ex}(x,t)\delta w(x,t)dx={\left\{\delta q(e)\right\}}^{T}\{f_{ex}\}$$

Using Hamilton’s principle, it can be gained
(46)$$\int_{t_{1}}^{t_{2}}\delta T_{e}-\delta U_{e}+\delta W_{e}dt=0$$
where $T_{e}$, $U_{e}$, and $W_{e}$ are the elemental kinetic energy, potential energy, and virtual work, respectively. From Equation (46), the dynamic motion equation for the ACLD pipeline in local coordinates can be written in the following form
(47)$$M^{e}{\ddot{q}}^{e}+C^{e}{\dot{q}}^{e}+K^{e}q^{e}=F^{e}$$

By using boundary conditions, the stiffness, mass, and damping element matrix are combined, from the local coordinate system to the global coordinate system. Finally, the dynamic equation of ACLD pipeline is obtained, and it can be represented as
(48)$$M\ddot{q}+C\dot{q}+Kq=F$$
where the assembled stiffness, mass, and damping matrices are K, M, and C, respectively. q is the displacement vector, and F is the external force vector. Considering the undamped system, under free vibration, the dynamic equation of ACLD system can be written as
(49)$$\left[K-\Omega^{2}M\right]\left\{q\right\}=0$$

From Equation (49), the natural frequencies ω and the loss factors η of the ACLD pipeline can be obtained.
(50)$$\Omega =\omega^{2}(1+i\eta )$$
(51)ω=(Re(Ω))12
(52)η=Im(Ω)Re(Ω)

## 3. Results and Discussion

### 3.1. Verification of Numerical Model

The ANSYS 19.1 finite element software (ANSYS, Canonsburg, PA, USA) package is used to model the ACLD pipeline. Due to the particularity of piezoelectric materials, a 3D solid element named SOLID5 is selected to model the piezoelectric layer. The element has eight nodes, and has four degrees of freedom at each node: UX, UY, UZ, VOLT. The base pipe and viscoelastic layer are modeled by a 3D solid element named SOLID 45. The element is defined by eight nodes and each node has three degrees of freedom: UX, UY, UZ. The adjacent components of the ACLD pipeline are shown in the Figure 3. The geometrical and material characteristics of each layer are plotted in Table 1.

This section discusses an example from Navin and Singh [26] to verify the finite element dynamic formula in the previous section. In their experiments, a cantilever beam that was partially covered with active and passive damping layers was studied. The materials and properties of each layers of the test are shown in Table 2.

In that test, the frequencies for an 80% covered PCLD beam were considered. The results were verified by the commercial software ANSYS (ANSYS 19.1, ANSYS, Canonsburg, PA, USA) and the numerical calculation software MATLAB (MATLAB, MathWorks, Natick, MA, USA), respectively. A comparison of the frequencies is shown in Table 3. In Table 4, the effect of the thickness variations of the VEM layer on the first mode loss factor for a 65% covered PCLD beam is shown.

From the above data, we can find that the authors’ results give good agreement with the reference results. Therefore, it can be concluded that the dynamic formula of the pipeline in the second chapter can give a correct analysis for the ACLD pipeline, and the presented model for pipeline with ACLD treatment is satisfactorily validated.

### 3.2. Parameterization of the Proposed Model

#### 3.2.1. The Influence of Viscoelastic Layer Parameters on Pipeline Vibration

Viscoelastic materials have good damping properties, which have the characteristics of both elastic materials and viscous materials. Therefore, viscoelastic materials can achieve energy conversion and dissipation in the form of mechanical damping when dynamic stress and strain are generated, to obtain the goal of vibration and noise reduction. Due to the inherent characteristics of viscoelastic materials, the viscoelastic layer plays a very important role in ACLD treatment. This part discusses the influence of the elasticity modulus and the thickness of the viscoelastic material on the natural frequencies and modal loss factors of the ACLD pipeline.

Figure 4 shows the influence of the elasticity modulus of the viscoelastic layer on frequencies and loss factors. It can be seen from the figure that the elasticity modulus of the viscoelastic layer has little influence on natural frequencies but a great effect on the loss factor. When the elasticity modulus ranges from 10^6^ to 10^7^, the modal loss factor of the first seven modes is improved, with a maximum for the sixth mode.

The effect of thickness of viscoelastic layer on natural frequency and loss factor is plotted in Figure 5. Since the change in the thickness of the viscoelastic layer has little effect on the quality and stiffness of a pipeline, it can be concluded that with the increase in the thickness of the viscoelastic layer, the change in natural frequencies of pipeline is not obvious. From the figure, it is evident that the influence of the thickness on the second mode loss factor is more remarkable than other modes. It is therefore found that the increasing thickness of the viscoelastic layer can improve the damping performance of the pipeline.

#### 3.2.2. Influence of ACLD Laying Circle Angle θ

When *x*1 = 0.1 m, the influence of ACLD circumference angle θ on natural frequencies and loss factors of pipeline is shown in Figure 6. From the curves, the ACLD circumference angle θ has a significant effect on the loss factors compared with the natural frequencies. For the damping properties, it is obvious that with the increase in θ, the loss factors increase dramatically. Therefore, in practical applications, a larger circumference coverage can be selected if the conditions permit.

#### 3.2.3. The Influence of the Length of ACLD

The effect of the length of the ACLD patch on the natural frequencies and loss factors is investigated with the thickness of the patch kept constant. When increasing the ACLD length from the middle of the pipe to both ends—that is, changing the value of *x*1—the variation of natural frequencies and loss factors of pipeline are plotted in Figure 7. From the curves, it is obvious that the length of ACLD patch has a significant effect on the loss factors of the pipeline. It is evident that the damping properties will increase with the increase in the length of ACLD patch.

When the total length of ACLD is kept constant at 0.3 m, there are three different laying methods:

(1) Lay a section of ACLD material in the middle of the pipeline in Figure 1; (2) as shown in Figure 8, the ACLD patch is evenly divided into two sections; and (3) as shown in Figure 9, the ACLD patch is evenly divided into three sections.

In Table 5, the natural frequencies and loss factors of pipeline in three cases are analyzed. Comparing the modal frequencies and loss factors of the pipeline in these three cases, it is found that in case 3, the pipeline has the best damping prosperities.

Taking the left end of the pipeline as the starting point, lay different lengths of ACLD patch to the right end, changing the value of L as shown in Figure 10. The effect of natural frequencies and loss factors on pipeline is illustrated in Figure 11. It can be obtained from the curve that the value of L has a more considerable effect on loss factors and the loss factors increase with the increase in the axial length of the ACLD patch.

#### 3.2.4. The Influence of Voltage

To verify the influence of the voltage value on the active constraining layer of pipeline vibration control, different control voltages are applied on the piezoelectric constraint layer, and the vibration reduction performance of the active restraint layer under different control voltages is compared. In this study, the constant acceleration amplitude excitation is used, and the acceleration a = 0.2 g. The acceleration response value of the middle node of pipeline is extracted. Figure 12 shows the results.

The results presented in Figure 12 show that with the increase in voltage, the acceleration response of pipeline first decreases and then increases. In other words, the damping effect of control voltage on pipeline first increases then decreases. The results of the amplitude of acceleration response under different control voltages are shown in Table 6. From the data, we can conclude that there is no significant difference in the droop rate when the control voltage are 50 and 100 V. Considering the economy and effectiveness, a control voltage of 50 V is the best choice.

#### 3.2.5. The Influence of Different Excitations

The vibration response of the pipeline under different excitations is analyzed in Figure 13. Different base excitation is applied to the pipeline, and the excitation of acceleration amplitudes are 0.1, 0.2, and 0.3 g, respectively. The applied voltage of the piezoelectric constraint layer is 50 V. From Figure 13, it can be seen that as the excitation amplitude increases, the peak value of the response of the pipeline increases. In other words, when the excitation amplitude is larger, the vibration of the pipeline becomes more intense.

## 4. Conclusions

In this paper, the parametric characteristics of the partially covered active constrained layer damping pipeline is discussed. The ACLD pipeline motion equation is derived via the developed finite element method. The finite element method is used to simulate the vibration of the ACLD pipeline, and the results show that the technology can obtain the damping effect of the pipeline. The influence of thickness and the elasticity modulus of viscoelastic layer, the length of ACLD patch, and the voltage applied on the piezoelectric layer is illustrated. The conclusions are as follows.

The best damping capability can be obtained by selecting the elasticity modulus and the thickness of viscoelastic layer. When the variation range of the elasticity modulus is [10^6^, 10^7^] Pa and the thickness is around 1 mm, the pipeline can get the best damping performance. With the circumferential coverage of ACLD increases, the modal loss factors of the pipeline increase steadily.

The length and the location of the ACLD patch have a remarkable influence on the vibration reduction of the pipeline system. Compared the loss factors, the length of ACLD has little effect on natural frequencies. A longer ACLD patch can improve the loss factors dramatically. When the length of ACLD remains unchanged, the number of ACLD divided equally and the location of the ACLD patch also have a great influence on the vibration reduction of the system.

The different voltages applied on the piezoelectric layer also have a significant effect on the vibration reduction of the pipeline. The response of the pipeline first decreases and then increases with the increase in voltage. In terms of economy and high efficiency, 50 V voltage is the best choice. The smaller the base excitation acceleration applied to the pipeline, the better the damping effect of the pipeline. The above conclusions could be used in the design and maintenance of pipeline systems for active vibration control.

## Figures and Tables

**Figure 1 materials-14-01209-f001:**
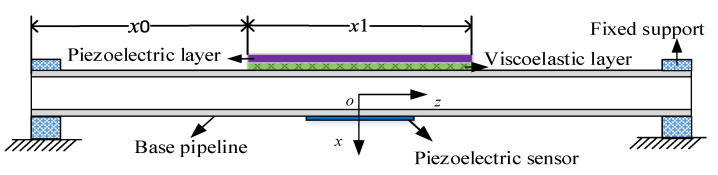
The fixed support model of pipeline.

**Figure 2 materials-14-01209-f002:**
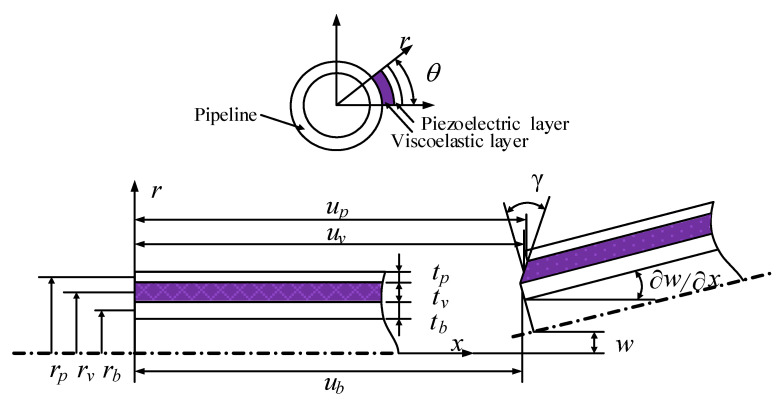
Deformation of the pipeline partially covered active constrained layer damping (ACLD).

**Figure 3 materials-14-01209-f003:**
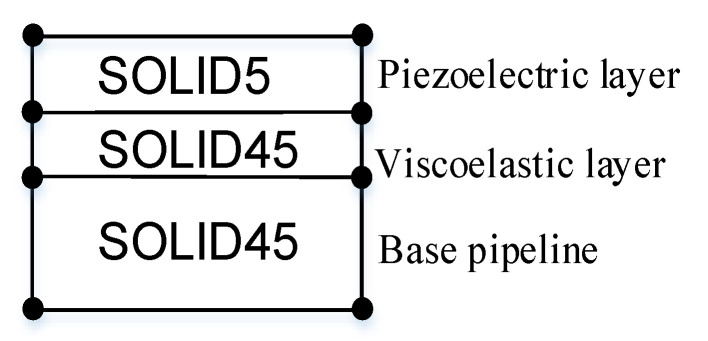
View of adjacent elements.

**Figure 4 materials-14-01209-f004:**
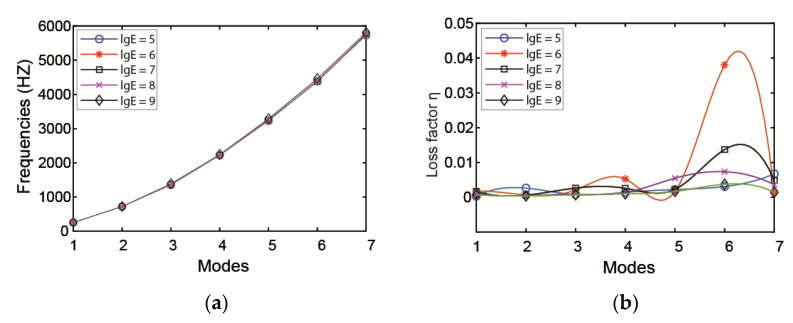
Influence of elasticity modulus of viscoelastic layer on (**a**) modal frequency and (**b**) loss factor.

**Figure 5 materials-14-01209-f005:**
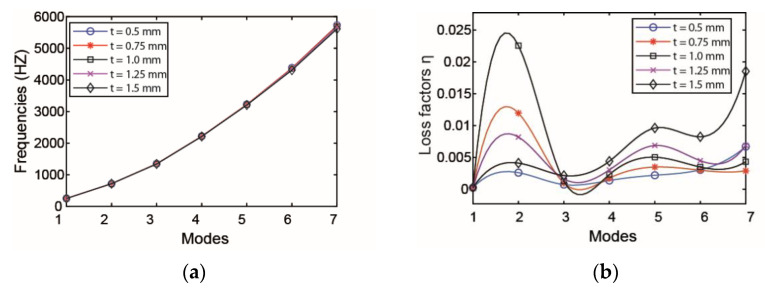
Influence of thickness of viscoelastic layer on the (**a**) modal frequency and (**b**) loss factor.

**Figure 6 materials-14-01209-f006:**
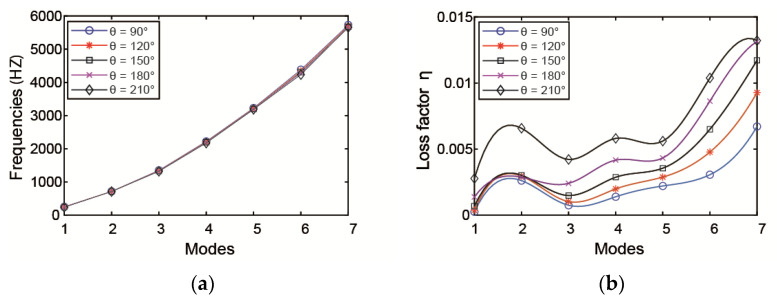
Influence of *θ* on the (**a**) modal frequency and (**b**) loss factor.

**Figure 7 materials-14-01209-f007:**
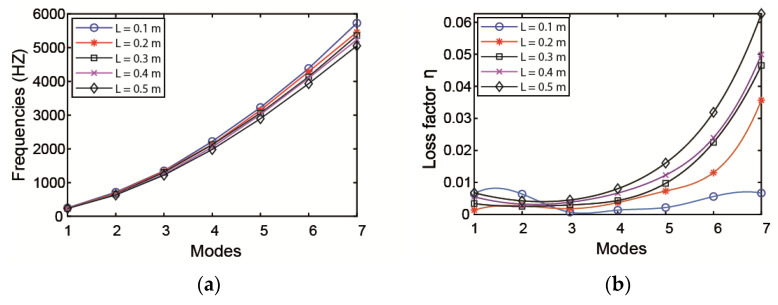
Influence of ACLD length on the (**a**) natural frequency and (**b**) loss factor.

**Figure 8 materials-14-01209-f008:**
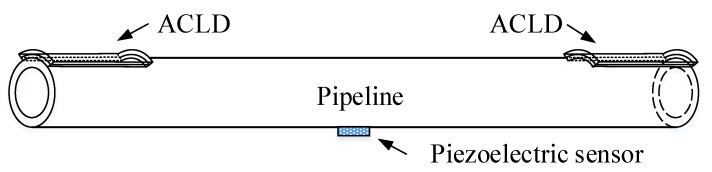
Schematic diagram of pipeline in case 2.

**Figure 9 materials-14-01209-f009:**
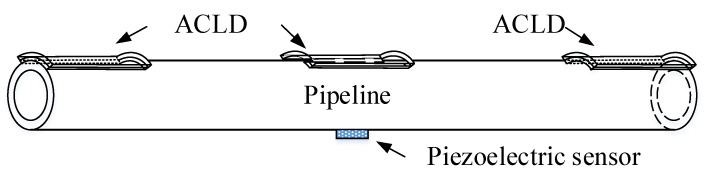
Schematic diagram of pipeline in case 3.

**Figure 10 materials-14-01209-f010:**
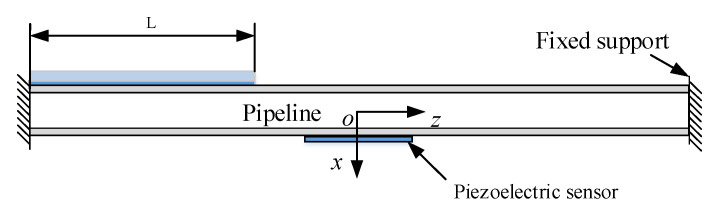
Axial length of ACLD on the pipeline.

**Figure 11 materials-14-01209-f011:**
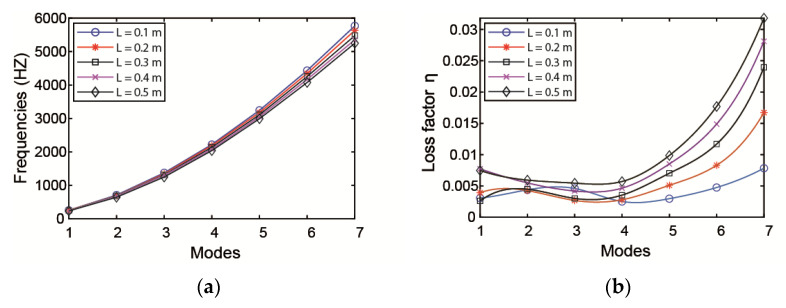
The influence of axial length of ACLD on the (**a**) natural frequency and (**b**) loss factor.

**Figure 12 materials-14-01209-f012:**
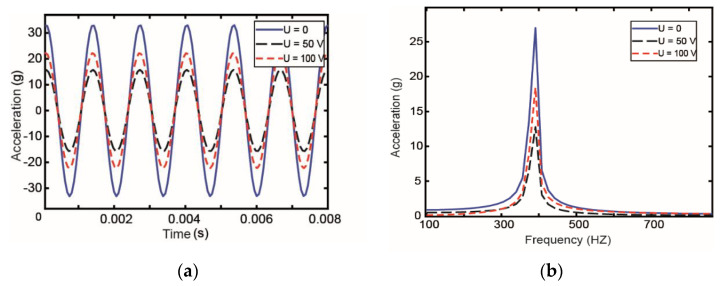
Vibration response in the (**a**) time domain and (**b**) frequency domain under different control voltages.

**Figure 13 materials-14-01209-f013:**
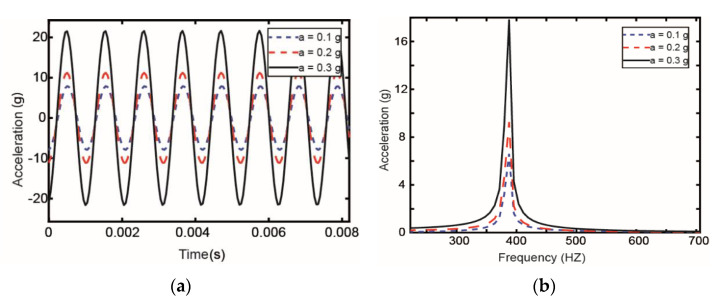
Vibration response in the (**a**) time domain and (**b**) frequency domain under different excitations.

**Table 1 materials-14-01209-t001:** Geometrical and material of ACLD pipeline.

Quantities	Base Layer	Viscoelastic Layer	Constraining Layer
Elastic modulus (Pa)	2.01 × 10^11^	4.5 × 10^5^	—
Density (kg/m^3^)	7860	980	7400
Thickness (mm)	1	0.5	1
Poisson ratio	0.3	0.5	0.3
Loss factor	—	0.9683	—
Radian	2π	π/2	π/2
Pipeline inner diameter = 7 mm; Length = 600 mm.			

**Table 2 materials-14-01209-t002:** The materials and properties of each layers.

Quantities	Base Layer	Viscoelastic Layer	Piezoelectric Layer
Elastic modulus (Pa)	7.1 × 10^10^	—	3 × 10^9^
Shear modulus (Pa)	—	5 × 10^7^(1 + 0.7i)	—
Density (kg/m^3^)	2700	1714	7500
Thickness (mm)	1.1	0.5–1.5	0.1
The length of the beam L = 200 mm; width b = 20 mm.			

**Table 3 materials-14-01209-t003:** The frequencies for an 80% covered passive constrained layer damping control (PCLD) beam.

Mode	1	2	3
Result in Reference [26] (HZ)	20.0	120.0	330.0
Result in ANSYS (HZ)	19.2	120.1	320.1
Result in MATLAB (HZ)	19.3	119.8	330.3

**Table 4 materials-14-01209-t004:** The first loss factor of coverage variations of the ACLD patch on the first mode for a 1.5 mm-thick viscoelastic material (VEM) layer.

Coverage (%)	65	80
Reference [26] loss factor	0.0363	0.0362
Present work loss factor	0.0364	0.0375
Error max (%)	2.7	3.5

**Table 5 materials-14-01209-t005:** Natural frequency and loss factor of pipeline in three cases.

	Frequency (HZ)			Loss Factor		
	Mode 1	Mode 2	Mode 3	Mode 1	Mode 2	Mode 3
Case 1	234.4	639.5	1252.8	5.34 × 10^−3^	3.11 × 10^−3^	3.05 × 10^−3^
Case 2	264.6	692.2	1336.5	2.64 × 10^−3^	3.46 × 10^−3^	4.05 × 10^−3^
Case 3	241.3	691.4	1321.8	1.25 × 10^−2^	4.65 × 10^−3^	1.46 × 10^−2^

**Table 6 materials-14-01209-t006:** The amplitude of acceleration response under different control voltages.

Control Voltages	Amplitude of Acceleration (g)	Droop Rate (%)
0 V	32.78	0
25 V	23.12	29.46
50 V	15.78	51.86
75 V	15.36	53.14
100 V	22.23	32.18

## Data Availability

The data presented in this study are available on request from the corresponding author.

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
