# Peer review of "Vibration and Damping Analysis of Pipeline System Based on Partially Piezoelectric Active Constrained Layer Damping Treatment"

_materials, 2021, doi:10.3390/ma14051209_

Round 1
Reviewer 1 Report
Typography Comments
===============
lines 32-34: revise sentence, repetitive use of "pipeline".
lines 45, 54, 56, 57, and every where in the paper: include a space between in-text reference citation and normal text.
lines 149, 165, 193, 203, 218, 237, 249: start with small letter "where" and not "Where".
lines 305-307: typo in Fig. 3 "SOLID..." and not "SOILD..."
line 316: mistake in Fig. 4 title, reference to shear modulus instead of elastic modulus.
lines 345-348: revise sentence, not clear to the reader.
Technical Comments
==================
lines 297-343: it is not clear the magnitude and type of base excitation used for all the analysis.
lines 307-328: it is redundant analysing the effect of elastic modulus change on vibration response. Such change would result in material configuration that are not viscoelastic. This section is not relevant. In this instance, it is more important the geometric changes considered in the paper .
lines 359-372: it is not clear the actual configuration of the pipe and active-passive constrained layer treatment used for this analysis from the various configurations discussed in the paper.
Reviewer 2 Report
I read the manuscript Vibration and damping analysis of pipeline system based on partially 1 piezoelectric active constrained layer damping treatment by authors Yuanlin Zhang et al., carefully and patiently. The topic of this manuscript is of interest to many researchers, readers of your valuable Journal. Without a doubt, the work is well written and organized - clear derivations, explanations, and figures. The English language is good. Also, I rebuilt most parts of the calculations and I found that they are correct.
I remarked on the originality and the clarity of the paper.
I suggest to the Editor-in-Chief accepting this manuscript, but after the authors make the following corrections:
- To being heightened more clearly the contributions of the authors in Introduction and Conclusions;
- After each relationship, a point or comma should be placed.
- Let a space on page 2 line 66 before [5] and on line 71 after [7].
- Write $t_niu$ in italic style in Eq. (6).
- The references are adequate, updated, and all they are necessary. Authors, please put DOI at the articles in References (for ex. [10]), as whereas you can. I think the authors must strengthen the References section with the titles that use the same technique, for instance:
- General solution in terms of complex potentials in antiplane states in prestressed and prepolarized piezoelectric crystals: application to Mode III fracture propagation, IMA J of Appl. Math. 70, 39-52, (2005). DOI:10.1093/imamat/hxh060;
- Creating the Coupled Band Gaps in Piezoelectric Composite Plates by Interconnected Electric Impedance. Materials, 11, 1656, (2018). DOI:3390/ma11091656.
I recommend the acceptance of this paper after Minor Revisions.
Reviewer 3 Report
Evaluation of the “Vibration and damping analysis of pipeline system based on partially piezoelectric active constrained layer damping treatment”
1Some phrases do not flow in the abstract “ The active control technology… loss factor have been discussed” Please ty a better formulation
2“ the pipelines and pipelines and the adjacent accessories”
3The scientific novelty of this work is limited, I suggest to rethink the novelty or present it in a 4better manner otherwise looks rather as an extension applied for pipeline vibration
5The overall English require massive improvements just an example “The fig.2 is plotted” which should read “The fig.2 plotted”
6You describe also some analytical calculation which were integrated in numerical model but you don’t have mentioned in abstract and or introduction
7You can define section 3. As “Results and discussion instead of “Comparison with the results in the literature” Probably you can but like 3.1. verification of numerical model
8Then you can put 3.2 Parameterization of the proposed model
9Table 1 is presented but not cited in text
10OK, about 3.1 but is not clear which numerical model you have used as in section 4. You present the software and numerical details so this is little bit ambiguity please restructure from section 3. Other wise not clear the meaning of section 3 and which boundary condition you have use
11The overall results are ok, but will be more powerful if they are validated and discussed against literature data as now there is no discussion
Round 2
Reviewer 3 Report
Thank you